# Room-temperature mechanocaloric effects in lithium-based superionic materials

Arun K. Sagotra[1], Dewei Chu[1] & Claudio Cazorla [1]

Mechanocaloric materials undergo sizable temperature changes during stress-induced phase transformations and hence are highly sought after for solid-state cooling applications. Most known mechanocaloric materials, however, operate at non-ambient temperatures and involve first-order structural transitions that pose practical cyclability issues. Here, we demonstrate large room-temperature mechanocaloric effects in the absence of any structural phase transformation in the fast-ion conductor $Li_3N$ ($|\Delta S| \sim 25$ J K$^{-1}$ kg$^{-1}$ and $|\Delta T| \sim 5$ K). Depending on whether the applied stress is hydrostatic or uniaxial the resulting caloric effect is either direct ($\Delta T > 0$) or inverse ($\Delta T < 0$). The dual caloric response of $Li_3N$ is due exclusively to stress-induced variations on its ionic conductivity, which entail large entropy and volume changes that are fully reversible. Our work should motivate the search of large and dual mechanocaloric effects in a wide variety of superionic materials already employed in electrochemical devices.

[1] School of Materials Science and Engineering, UNSW Sydney, Sydney, NSW 2052, Australia. Correspondence and requests for materials should be addressed to C.C. (email: c.cazorla@unsw.edu.au)

Caloric materials undergo large temperature variations as a result of field-induced phase transformations involving sizable entropy changes[1]. Solid-state cooling based on caloric materials represents an alternative to traditional refrigeration technologies based on compression cycles of greenhouse gases, which in addition to the obvious environmental threats cannot be scaled to small sizes. A major breakthrough took place more than two decades ago with the discovery of giant magnetocaloric effects in the intermetallic compound $Gd_5(Si_2Ge_2)$[2]. Magnetocaloric materials, however, are mostly based on rare-earth elements, thereby are scarce, and require the generation of ultrahigh magnetic fields. Meanwhile, mechanocaloric (MC) effects produced by moderate mechanical stresses recently have attracted a lot of attention owing to their large isothermal entropy and adiabatic temperature shifts, which may surpass by an order of magnitude those achieved in magnetocaloric materials with commercially available permanent magnets[1,3]. To date, giant MC effects near structural phase transitions have been experimentally observed in a number of relatively expensive magnetic materials[4–6], shape-memory alloys[7,8], fluoride-based materials[9–12], polar materials[13,14], a polymer[15], organic—inorganic hybrid perovskites[16,17], and the archetypal superionic conductor AgI[18]. We note that giant MC effects in AgI and other similar superionic compounds like $CaF_2$, $PbF_2$, and $Li_3OCl$ were theoretically predicted prior to the experiments[19,20]. Actually, computational approaches are increasingly being applied with success to the study of MC materials (see, for instance, refs.[21,22]) since they are reliable and can be used in an effective and inexpensive manner. Yet, in spite of their promise, giant MC effects normally occur at temperatures that are distant from ambient conditions and involve first-order transitions that in practice lead to severe thermal hysteresis problems. New caloric materials displaying large room-temperature MC effects and non-irreversible atomic processes, therefore, are highly desirable for the development of solid-state cooling devices.

We present here an original strategy for achieving large and reversible MC effects at ambient temperature based on fast-ion conductors, and illustrate it in $Li_3N$. Rather than focusing on the triggering of a structural phase transition, which only serendipitously will occur at room temperature, our starting point is a material that is already superionic at ambient conditions. Large and reversible MC effects then can be obtained through the application of mechanical stresses since these affect significantly and sustainedly the ionic conductivity, and in turn the entropy and volume, in fast-ion conductors[19,20,23]. (We note that application of the same strategy to other families of MC materials, for instance ferro/ferrielectrics and magnetic compounds, is likely to fail as due to field-induced saturation of the relevant order parameters at thermodynamic conditions other than transition points.) In $Li_3N$, we estimate a giant room-temperature isothermal entropy change $\Delta S$ of about $-25$ J K$^{-1}$ kg$^{-1}$ at a maximum hydrostatic stress of 1 GPa. The corresponding room-temperature adiabatic temperature shift $\Delta T$ is $\approx +3$ K, which is large but not giant due to the huge heat capacity of such a light-weight material. Interestingly, when $Li_3N$ is subjected to uniaxial tensile loads similarly large $|\Delta S|$ and $|\Delta T|$ shifts are obtained but with opposite signs ($\Delta S > 0$). The disclosed large and dual, that is, direct ($\Delta T > 0$) and inverse ($\Delta T < 0$), MC effects can be understood in terms of simple arguments based on stress-induced variations on the volume an ion-transport properties of superionic materials[19,20,23]. We also discuss straightforward chemical strategies for increasing the $\Delta T$ values estimated in $Li_3N$ by more than a factor of two.

## Results

**Direct barocaloric effects.** Figure 1 shows the two common polymorphs found in commercially available samples of bulk $Li_3N$. The $\alpha$ phase (hexagonal, space group $P6_3/mmm$) has a layered structure composed of alternating planes of hexagonal $Li_2N$ and pure Li$^+$ ions (Fig. 1a). The $\beta$ phase (hexagonal, space group $P6_3/mmc$) presents an additional layer of lithium ions intercalated between the $Li_2N$ planes that is accompanied by a doubling of the unit cell (Fig. 1b). Exceptionally high ionic conductivities of the order of $10^{-4}$–$10^{-3}$ S cm$^{-1}$ have been experimentally observed in $Li_3N$ at room temperature[24–26]. In our molecular dynamics (MD) simulations based on classical interaction potentials and first-principles methods (Methods, Supplementary Methods, and Supplementary Table 1), we find that stoichiometric $\alpha$-$Li_3N$ displays high ionic conductivity at $T = 300$ K (i.e., of the order of $10^{-4}$ S cm$^{-1}$), which is in consistent agreement with the experimental observations[24–26]. On the other hand, it is necessary to consider a small concentration of extrinsic Li$^+$ vacancies (~1%) in $\beta$-$Li_3N$ to render similar room-temperature superionic features.

Figure 2 shows our estimation of the barocaloric (BC) effects induced by hydrostatic stresses of up to 1 GPa in $Li_3N$ at room temperature and $T = 400$ K. In the $\beta$ phase, we obtain a maximum room-temperature entropy shift of $-24$ J K$^{-1}$ kg$^{-1}$ that increases slightly in absolute value at higher temperatures (Fig. 2a). The resulting BC effects, therefore, are direct ($\Delta T > 0$, Methods). In view of previous reports, see for instance[4–6] and Supplementary Table 2, the magnitude of such $\Delta S$ can be regarded as "giant" (that is, $|\Delta S| \sim 10^1$ J K$^{-1}$ kg$^{-1}$). The accompanying adiabatic temperature shifts (Fig. 2b), however, are large but not giant (that is, $|\Delta T| < 10$ K) owing to the huge heat capacity of $Li_3N$ ($C_0 \sim 4.10^3$ J K$^{-1}$ kg$^{-1}$ at $T = 300$ K, Supplementary Fig. 1), which is a very light-weight compound. We note that for a fixed temperature the pressure-induced increase of $\Delta S$ and $\Delta T$ are practically constant, which suggests a sustained change in the physical properties of the material under pressure (in contrast to the abrupt $\Delta S$ and $\Delta T$ changes observed during structural phase transitions[20]). Analogous results are found in $\alpha$-$Li_3N$ (Fig. 2c, d) although in this case the estimated isothermal entropy and adabatic temperature changes are slightly larger (for instance, $\Delta S = -32$ J K$^{-1}$ kg$^{-1}$ and $\Delta T = +2.8$ K at

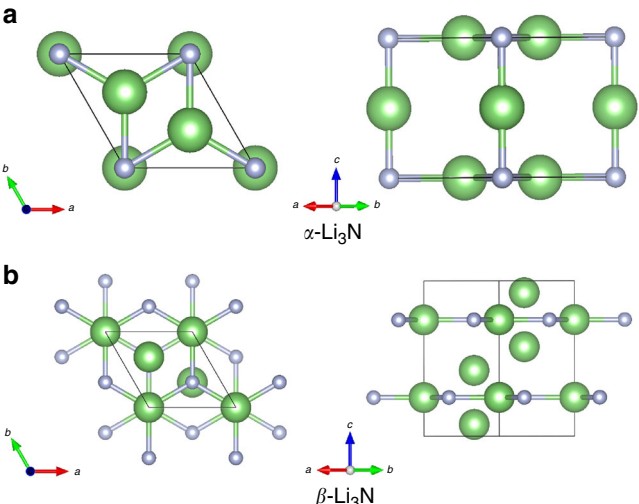

**Fig. 1** Representation of the $Li_3N$ polymorphs considered in this study. **a** Sketch of the $\alpha$ phase from different views. **b** Sketch of the $\beta$ phase from different views. Li and N ions are represented with green and blue spheres, respectively

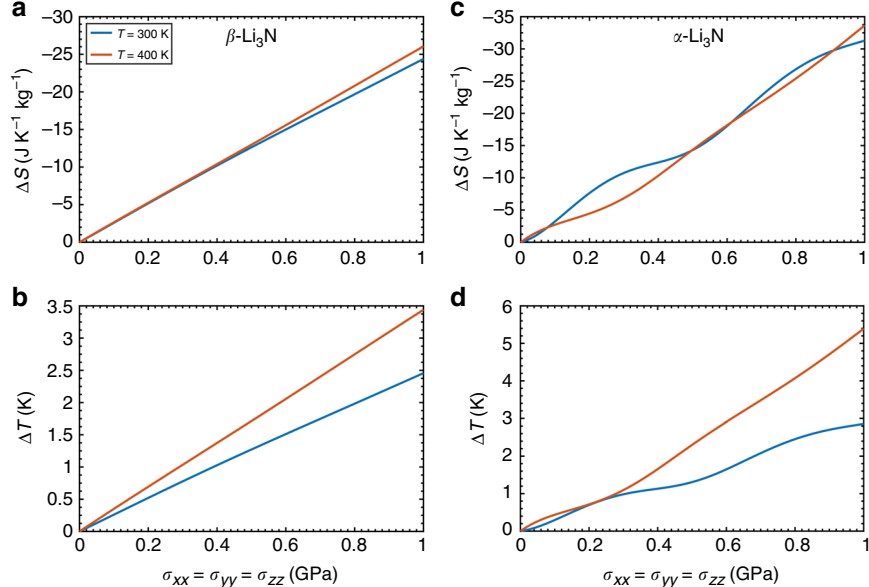

**Fig. 2** Direct barocaloric effects in Li₃N. **a** Isothermal entropy and **b** adiabatic temperature shifts estimated in $\beta$-Li₃N. **c** Isothermal entropy and **d** adiabatic temperature shifts estimated in $\alpha$-Li₃N

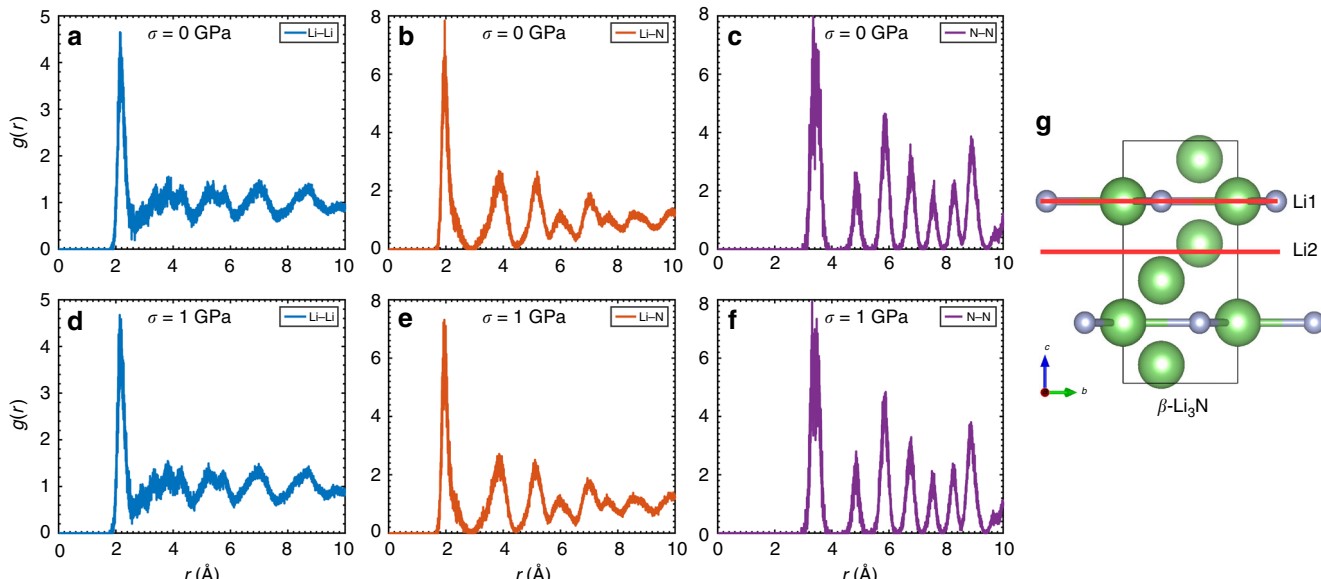

**Fig. 3** Structural properties of $\beta$-Li₃N at equilibrium and under pressure. Radial pair distribution function obtained for the **a**, **d** Li–Li, **b**, **e** Li–N, and **c**, **f** N–N pairs at $T = 300$ K. **g** Two planes perpendicular to the hexagonal c-axis and containing different types of Li⁺ ions, namely "Li1" and "Li2", are indicated with red thick lines

$T = 300$ K and $\sigma = 1$ GPa), and the corresponding pressure-induced shift increments fluctuate somewhat.

The large and direct BC effects found in superionic Li₃N are not driven by any structural phase transformation (i.e., the crystal symmetry of the system remains the same over the whole studied pressure range). This is explicitly shown in Fig. 3a–f, where we plot the radial distribution function calculated for all ionic couples at $T = 300$ K and different $\sigma$ conditions: The corresponding $g(r)$ profiles present almost identical traits independently of the applied pressure within the interval $0 \leq P \leq 1$ GPa. Further evidence for the absence of any $\sigma$-induced structural phase transformation is provided by our coordination number (Supplementary Fig. 2), position correlation function[27,28] (Supplementary Fig. 3), and enthalpy energy (Supplementary Fig. 4)

results. We note that our findings are fully consistent with previous experimental as well as theoretical reports[29–31], which provides further confidence in the employed computational approach (Methods and Supplementary Methods).

The absence of any structural phase transformation in Li₃N (within the pressure range $P \leq 1$ GPa) is in stark contrast to what is observed in other BC materials, in which most of the caloric response is concentrated near phase transition points[1]. Then, which is the principal mechanism behind the disclosed giant $\Delta S$ in Li₃N? It is well-known that hydrostatic pressure depletes significantly the ionic diffusivity, and therefore the entropy, in most fast-ion conductors[19,27,28,32]. Essentially, the available volume to interstitial ions is effectively reduced under compression and as a consequence the kinetic barriers and formation

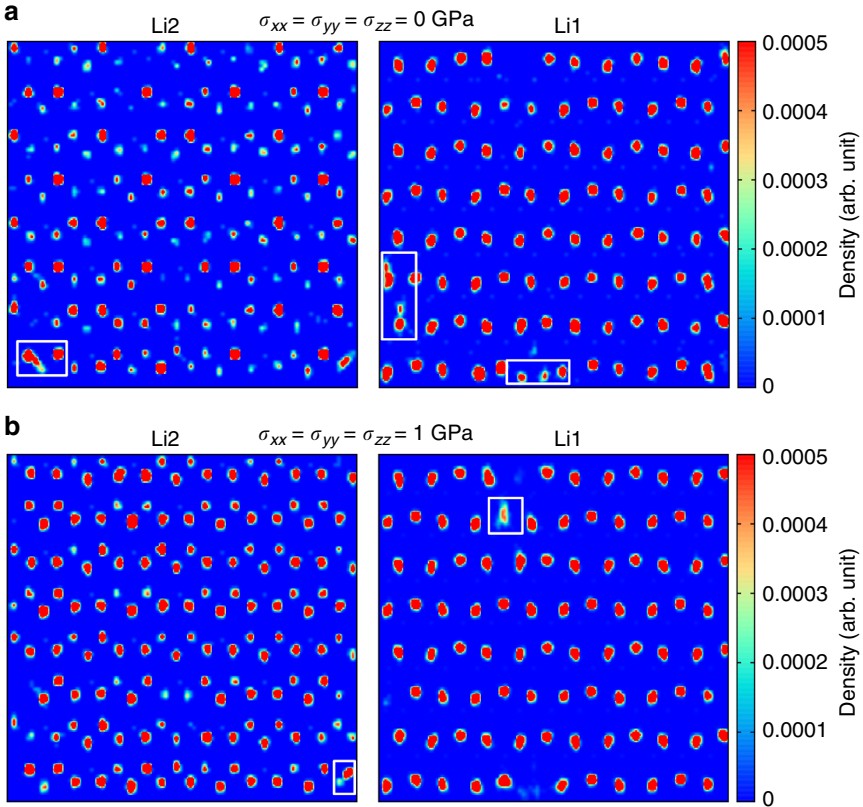

**Fig. 4** Lithium ion density plots calculated in $\beta$-Li$_3$N at $T = 300$ K and different hydrostatic pressure conditions. Results obtained at **a** zero pressure and **b** $\sigma = 1$ GPa. "Li1" and "Li2" indicate results obtained for system slices centered in the $z$-positions sketched in Fig. 3g. White boxes emphasize Li$^+$ diffusion within the $x$–$y$ plane

energy of defects governing ion migration increase drastically[19,28] (Supplementary Fig. 5). We illustrate such a $\sigma$-induced reduction in the ionic conductivity of $\beta$ – Li$_3$N in Fig. 4. Therein, we plot two different lithium density plots constructed as in-plane Li$^+$ position histograms (see Fig. 3g for plane notation) calculated at room temperature and $\sigma = 0$ and 1 GPa: The diffusivity of lithium ions at equilibrium (Fig. 4a) is larger than under pressure (Fig. 4b) as shown by the higher presence of high-density areas between crystal lattice sites and on vacancy positions (highlighted with white boxes in the figure). Specifically, the estimated room-temperature Li$^+$ diffusion coefficient $D_{Li}$ and corresponding ionic conductivity $\phi_{Li}$ (Methods) amount respectively to $8.6 \times 10^{-7}$ cm$^2$ s$^{-1}$ and $4.7 \times 10^{-4}$ S cm$^{-1}$ at $\sigma = 0$ GPa, and to $5.3 \times 10^{-7}$ cm$^2$ s$^{-1}$ and $2.8 \times 10^{-4}$ S cm$^{-1}$ at $\sigma = 1$ GPa (Supplementary Fig. 6). We note that in our ab initio and classical MD simulations the Li$^+$ diffusivities calculated along the hexagonal $c$–axis and other perpendicular directions do not present significant differences (Supplementary Fig. 7).

**Inverse elastocaloric effects.** Uniaxial and biaxial mechanical stresses can be used to enhance the ionic conductivity in superionic materials[19,20,23,33–35]. An enhancement of ionic conductivity normally is accompanied by an increase in the entropy of the system ($\Delta S > 0$), which suggests the possibility of realizing also inverse MC effects ($\Delta T < 0$) in fast-ion conductors. We have tested such an hypothesis in Li$_3$N by performing MD simulations under constrained uniaxial tensile loads. (We note that $\alpha$–Li$_3$N is found to be vibrationally unstable in our uniaxial tensile stress simulations, hence we focus on the $\beta$–Li$_3$N polymorph in the following analysis.)

Figure 5 shows the elastocaloric (EC) effects induced by uniaxial tensile stresses applied along the hexagonal $c$-axis in $\beta$-Li$_3$N at room temperature and $T = 400$ K. An arbitrary maximum tensile stress of $-8$ GPa has been applied in our calculations, which corresponds to a maximum strain deformation along the hexagonal $c$-axis of $+5.8\%$ (see inset in Fig. 5). A giant isothermal entropy change of $\Delta S = +20$ J K$^{-1}$ kg$^{-1}$ is estimated at the highest simulated uniaxial load and room temperature, which increases up to $+25$ J K$^{-1}$ kg$^{-1}$ at $T = 400$ K (Fig. 5a). The resulting EC effects, therefore, are inverse ($\Delta T < 0$). Also in this case, the accompanying adiabatic temperature shifts are large but not giant (that is, $\Delta T = -2.0$ and $-3.4$ K at $T = 300$ and 400 K, respectively, Fig. 5b) due to the huge heat capacity of Li$_3$N (Supplementary Fig. 1).

In analogy to the BC case, none $\sigma$-induced structural phase transition is responsible for the observed EC effects (Supplementary Figs. 2, 4, 8). The inverse EC response of superionic Li$_3$N can be entirely understood in terms of $\sigma$-induced enhancements of its ionic conductivity. Uniaxial tensile stresses produce an effective increase in the volume of the system, which in turn provokes the kinetic barriers and formation energy of defects governing ion migration to decrease significantly[19,34,35] (Supplementary Fig. 5). Such a $\sigma$-driven enhancement of the Li$^+$ ionic conductivity is illustrated in Fig. 6, where we show that the number of high-density lithium ion regions appearing between crystal lattice positions and on vacancy sites is appreciably higher than found at $\sigma = 0$ GPa and same temperature (Fig. 4a). Likewise, for an uniaxial tensile stress of $-7$ GPa the estimated room-temperature lithium diffusion coefficient $D_{Li}$ and corresponding ionic conductivity $\phi_{Li}$ (Methods) amount to $10.2 \times 10^{-7}$ cm$^2$ s$^{-1}$ and $5.5 \times 10^{-4}$ S cm$^{-1}$, respectively, to be compared with $8.6 \times 10^{-7}$

cm$^2$ s$^{-1}$ and 4.7×10$^{-4}$ S cm$^{-1}$ calculated at $T = 300$ K and $\sigma = 0$ GPa (Supplementary Fig. 6).

## Discussion

The large MC room-temperature effects disclosed in superionic Li$_3$N pose great prospects in the context of refrigeration-cycle reversibility as due to the absence of mechanical hysteresis effects deriving from the nucleation of order-parameter domains (in contrast to what occurs in ferro/ferrielectrics and magnetic compounds where such issues may actually turn out to be

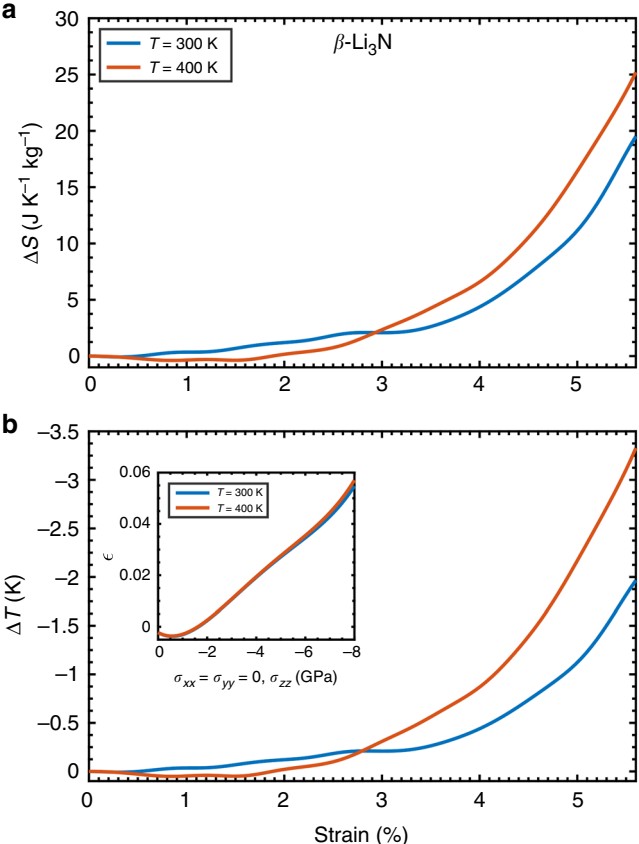

**Fig. 5** Inverse elastocaloric effects in $\beta$-Li$_3$N. **a** Isothermal entropy and **b** adiabatic temperature shift results. The relationship between uniaxial tensile stress and strain deformation along the Cartesian $z$ direction, or equivalently the hexagonal $c$-axis, is shown in the inset

critical). Furthermore, it has been experimentally demonstrated by means of ultrafast X-ray spectroscopy that the characteristic time scale of superionic switching is just of few picoseconds[36], which is consistent with our simulations (Supplementary Fig. 7). Consequently, refrigeration cycles based on fast-ion conductors may be conducted efficiently at high repetition rates. In order to assess the reversibility features of superionic-based MC cycles, we have performed additional classical MD simulations in which we have varied at a constant rate of 10$^{-3}$ GPa/ps the applied pressure on $\beta$-Li$_3$N, first from 0 to 1 GPa and then backwards from 1 to 0 GPa, while monitoring the accompanying changes on the volume and energy of the system. Our simulation results reassure the highly reversible and responsive nature of $\sigma$-driven changes on the physical properties of Li$_3$N, as it is demonstrated by the smallness of the volume and energy differences estimated between the 0→1 GPa and 1→0 GPa runs (Supplementary Fig. 9).

Arguably, the only disadvantage of superionic Li$_3$N for developing new MC solid-sate cooling applications is that the corresponding adiabatic temperature shifts are not giant ($|\Delta T| < 10$ K). As we have mentioned in previous sections, this shortcoming is due to the fact that the heat capacity of Li$_3$N $C_0$ is huge owing to its light weight (note that giant isothermal entropy changes are already obtained under moderate stress variations, see Eq. (5) in Methods). A straightforward strategy for drastically enhancing the $\Delta T$ values obtained in Li$_3$N may consist in substituting the "static" sublattice of nitrogen ions by isoelectronic and larger-mass larger-radius elements, namely, P and As, in order to effectively reduce the $C_0$ of the system while keeping its super-ionicity[28]. Certainly, Li$_3$P and Li$_3$As are isostructural to Li$_3$N and they are known to be better superionic conductors at room temperature[26]. Consequently, one can reasonably expect that the $\sigma$-induced $\Delta S$ shifts appearing in Li$_3$P and Li$_3$As will be at least as large as estimated in Li$_3$N[37], and then trivially estimate a lower bound for the increase in cooling performance due to their decrease in heat capacity. We have performed first-principles quasi-harmonic calculations of the heat capacity (Methods) in Li$_3$P and Li$_3$As as a function of temperature (Supplementary Fig. 1), and found that at $T = 300$ K their $C_0$'s are respectively 1.3 and 2.2 times smaller than the estimated in Li$_3$N. In addition, the $T$-induced volume expansion $\left(\frac{\partial V}{\partial T}\right)_\sigma$ in Li$_3$As is about 1.4 times larger than the estimated in Li$_3$N at room temperature (Supplementary Fig. 10). These results, namely, smaller $C_0$'s and larger $\left(\frac{\partial V}{\partial T}\right)_\sigma$'s, suggest that giant adiabatic temperature changes ($|\Delta T| > 10$ K) actually may be realized in lithium-based fast-ion conductors (see Eqs. (3) and (5) in Methods).

Our MC findings on Li$_3$N should stimulate the development of room-temperature cooling devices based on fast-ion conductors, whose energy efficiency and refrigerant performance are good as

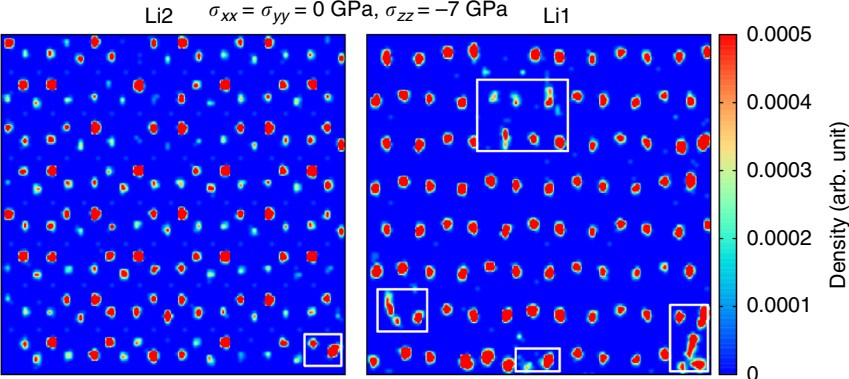

**Fig. 6** Lithium ion density plots calculated in $\beta$-Li$_3$N at $T = 300$ K under an uniaxial tensile stress of $-7$ GPa along the Cartesian $z$ direction. "Li1" and "Li2" indicate results obtained for system slices centered in the $z$-positions sketched in Fig. 3g. White boxes emphasize Li$^+$ diffusion within the $x$–$y$ plane

compared to magneto and electrocaloric materials. Unlike other caloric materials, the MC effects revealed in $Li_3N$ are not driven by any $\sigma$-induced structural phase transition but instead by changes on its ionic conductivity. As a result either direct or inverse reversible MC effects can be achieved at room temperature, just depending on how the mechanical stresses are applied. Biaxial stresses are likely to render similar MC effects than obtained with uniaxial stresses, adding the particularity that compressive loads also can be accomplished[20] (Supplementary Fig. 11). The original strategy introduced in this study for realizing large and reversible MC effects at room temperature is not exclusive of lithium-based fast-ion conductors, it should work equally well for any other family of materials exhibiting super-ionicity and robust structural stability at ambient conditions (e.g., solid oxide fuel cell materials).

## Methods

**Classical molecular dynamics simulations**. MD ($N,P,T$) simulations are performed with the LAMMPS code[38]. The pressure and temperature in the system are kept fluctuating around a set-point value by using thermostatting and barostatting techniques in which some dynamic variables are coupled to the particle velocities and simulation box dimensions. The interactions between atoms are modeled with rigid-ion Born-Mayer-Huggins potentials[39]. We employ large simulation boxes, typically containing up to 8000 atoms, and apply periodic boundary conditions along the three Cartesian directions. Defective $\beta$-$Li_3N$ systems are generated by randomly removing a specific number of cations and anions from the system in order to reproduce the experimentally observed ionic conductivity data[24] and to maintain the condition of charge neutrality. Newton's equations of motion are integrated using the customary Verlet's algorithm with a time-step length of $10^{-3}$ ps. The typical duration of a MD run is of 200 ps. A particle–particle particle–mesh $k$-space solver is used to compute long-range van der Waals and Coulomb interactions beyond a cut-off distance of 12 Å at each time step. We explicitly have checked that by creating point defects in an ordered manner and/or by increasing the total duration of the MD simulations up to 500 ps, our results remained invariant within the statistical uncertainties[23]. Further details of our classical MD simulations (e.g., interatomic potential models and parameters) can be found in the Supplementary Methods and Supplementary Table 1.

**Density functional theory calculations**. First-principles calculations based on density functional theory (DFT) are performed to analyze the energy, structural, vibrational, and ionic transport properties of $Li_3N$. We perform these calculations with the VASP code[40] by following the generalized gradient approximation to the exchange-correlation energy due to Perdew et al.[41]. Possible dispersion interactions are captured with the D3 correction scheme developed by Grimme and co-workers[42]. The projector augmented-wave method is used to represent the ionic cores[43], and the electronic states $1s$-$2s$ of Li and $2s$-$2p$ of N are considered as valence. Wave functions are represented in a plane-wave basis truncated at 650 eV. By using these parameters and dense $\mathbf{k}$-point grids for Brillouin zone integration, the resulting energies are converged to within 1 meV per formula unit. In the geometry relaxations, a tolerance of 0.01 eV·Å$^{-1}$ is imposed in the atomic forces. Ab initio MD (AIMD) simulations are carried out to assess the reliability of the interatomic potential models employed in the classical MD simulations. Details of our AIMD tests can be found in the Supplementary Methods.

We also perform ab initio phonon frequency calculations with the small-displacement method in order to assess the vibrational stability of the analyzed systems and estimate their (vibrational) heat capacity. In the small-displacement method the force-constant matrix is calculated in real-space by considering the proportionality between atomic displacements and forces[44–46]. The quantities with respect to which our phonon calculations are converged include the size of the supercell, the size of the atomic displacements, and the numerical accuracy in the sampling of the Brillouin zone. We find the following settings to provide quasi-harmonic free energies converged to within 5 meV per formula unit[46]: $3 \times 3 \times 3$ supercells (where the figures indicate the number of replicas of the unit cell along the corresponding lattice vectors), atomic displacements of 0.02 Å, and $\mathbf{q}$-point grids of $14 \times 14 \times 14$. The value of the phonon frequencies are obtained with the PHON code developed by Alfè[45]. In using this code we exploit the translational invariance of the system, to impose the three acoustic branches to be exactly zero at the center of the Brillouin zone, and apply central differences in the atomic forces.

Finally, we perform ab initio nudged-elastic band (NEB) calculations[47] to estimate the kinetic energy barriers for lithium ion diffusion in $Li_3N$ under different hydrostatic pressure conditions (Supplementary Fig. 5). The calculations are performed in a $2 \times 2 \times 1$ supercell containing 32 atoms. We use a $\mathbf{q}$-point grid of $7 \times 7 \times 10$ and an energy plane-wave cut-off of 650 eV. Six intermediate images are used in our NEB calculations and the geometry optimizations are finalized once the total forces on the atoms are smaller than 0.01 eV·Å$^{-1}$.

**Estimation of key quantities**. Ionic diffusion coefficients are calculated with the formula:

$$D_i = \lim_{t \to \infty} \frac{\langle |R_i(t + t_0) - R_i(t_0)|^2 \rangle}{6t}, \tag{1}$$

where $R_i(t)$ is the position of the migrating ion labeled as $i$ at time $t$, $t_0$ an arbitrary time origin, and $\langle \cdots \rangle$ denotes average over time and particles. The mean squared displacement of each ionic species is defined as $\langle \Delta R_i^2(t) \rangle \equiv \langle |R_i(t + t_0) - R_i(t_0)|^2 \rangle$. Likewise, ionic conductivities are obtained with the well-known Nernst-Einstein relationship[48]:

$$\phi_i = \frac{Ne^2 D_i}{k_B T}, \tag{2}$$

where $N$ and $e$ are the number density and charge of $Li^+$ ions, respectively.

Isothermal entropy changes associated to the BC effect are estimated as[1]:

$$\Delta S(\sigma_f, T) = -\int_0^{\sigma_f} \left(\frac{\partial V}{\partial T}\right)_\sigma d\sigma, \tag{3}$$

where $\sigma_f$ represents the applied hydrostatic pressure, and $V$ the volume of the system. In the case of EC effects, the same quantity is calculated as[1]:

$$\Delta S(\sigma_f, T) = V_0 \cdot \int_0^{\sigma_f} \left(\frac{\partial \varepsilon}{\partial T}\right)_\sigma d\sigma, \tag{4}$$

where $\sigma_f$ now represents the applied mechanical stress along the Cartesian $z$ direction, $\varepsilon$ the strain deformation that the system undergoes along the same direction (i.e., $\varepsilon(\sigma, T) \equiv \frac{L_z(\sigma,T) - L_z(0,T)}{L_z(0,T)}$ where $L_z$ corresponds to the length of the simulation box along the Cartesian $z$ direction), and $V_0$ the equilibrium volume of the system. Finally, the resulting adiabatic temperature shifts are estimated with the formula:

$$\Delta T(\sigma_f, T) = -\frac{T}{C_0(T)} \cdot \Delta S(\sigma_f, T), \tag{5}$$

where $C_0(T) = \left(\frac{dU}{dT}\right)_{V_0}$ is the heat capacity of the crystal calculated at zero-stress conditions. Within the quasi-harmonic approximation[44–46], this latter quantity can be calculated as:

$$C_0(T) = \frac{1}{N_q} \sum_{\mathbf{q}s} \frac{\left(\hbar\omega_{\mathbf{q}s}\right)^2}{k_B T^2} \times \frac{e^{\frac{\hbar\omega_{\mathbf{q}s}}{k_B T}}}{\left(e^{\frac{\hbar\omega_{\mathbf{q}s}}{k_B T}} - 1\right)^2}, \tag{6}$$

where $N_q$ is the total number of wave vectors used for integration in the Brillouin zone, the summation runs over all wave vectors $\mathbf{q}$ and phonon branches $s$, and $\omega_{\mathbf{q}s}$ are the phonon frequencies of the material at the equilibrium volume. Further technical details on our calculations can be found in the Supplementary Methods.

**Data Availability**. The data that support the findings of this study are available from the corresponding author (C.C.) upon reasonable request.

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

## Acknowledgements

This research was supported under the Australian Research Council's Future Fellowship funding scheme (No. FT140100135). Computational resources and technical assistance were provided by the Australian Government and the Government of Western Australia through Magnus under the National Computational Merit Allocation Scheme and The Pawsey Supercomputing Centre.

## Author contributions

A.K.S., D.C., and C.C. conceived the study and planned the research. A.K.S. and C.C. performed the theoretical calculations. Results were discussed by A.K.S., D.C., and C.C. The manuscript was written by A.K.S. and C.C. with substantial input from D.C.

## Additional information

**Competing interests:** The authors declare no competing interests.

