## [Peer Review File · Nature Communications]

Reviewers' comments:

Reviewer #1 (Remarks to the Author):

The submitted paper describes a room-temperature mechanocaloric effects in lithium-based superionic materials. The mechanocaloric effects ($|\Delta S| \sim 25 \text{ JK}^{-1}\text{kg}^{-1}$ at a maximum hydrostatic stress of 1 GPa and $|\Delta T| \sim 5 \text{ K}$) of Li_3N are not remarkable compared with the rare-earth serial materials, however Li_3N has a distinctly different feature. As the fast-ion conductor, Li_3N has no structural phase transformation, so there is no hysteresis in the phase transformation which is fully reversible. This topic is interesting, and the presentation of the results appears to be clear and the paper fairly well written. Therefore, I recommend the paper should be published.

Reviewer #2 (Remarks to the Author):

The authors have reported a so called large mechanocaloric effects in the fast-ion conductor Li_3N which is absence of any structural phase transformation. The results sound interesting, but there are several issues still unclear.

1. The description of the measurement of the mechanocaloric effect in the manuscript make me confusing which is the base of the manuscript. I. e., the reliability of the temperature change, and the durable.
2. The pressure and temperature dependence of T_{ad} and ΔS should be provided.
3. The origion of such large MC effect should be discussed.
4. The discussion related to difference and relation between the materials with and without structure transformation are suggested.
5. Additionally, I do not really agree with the authors declared that, the sample is without structure transformation, such high pressure may at least induce the lattice distortion, it is better to provide more evidence.

Reviewer #3 (Remarks to the Author):

In the manuscript entitled "Room-temperature mechanocaloric effects in lithium-based superionic materials", the authors present promising theoretical simulations on giant barocaloric effects in the Li_3N superionic materials, specifically in the α - Li_3N and β - Li_3N polymorphs, at 300 and 400 K. This is a very interesting and well-written work on the growing topic of mechanocaloric materials. The more striking features of the here exposed manuscript is the calculation of giant mechanocaloric effects in the absence of any structural transition found in two Li_3N polymorphs. Nevertheless, I found several major concerns that authors should address previous publication of this work:

(1) According to the authors, the giant mechanocaloric effects showed by α - Li_3N and β - Li_3N polymorphs are not driven by any structural transition. Nevertheless, there are several publications reporting several pressure-induced phase transitions in Li_3N [S. Cui et al., *Solid State Commun.*, 2009, 149, 612; Y. Yan et al., *Eur. Phys. J. B*, 2008, 61, 397; J. C. Schön et al., *J. Mater. Chem.*, 2001, 11, 69; M. Mali et al., *Phys. Rev. B*, 1987, 36 (7), 3888; A. C. Ho et al., *Phys. Rev. B*, 1999, 59 (9), 6083]. Even more, it is reported that at around 0.42 GPa there is a pressure-induced phase transition from α - Li_3N to β - Li_3N [W. Li et al., *Energy Environ. Sci.*, 2010, 3, 1524], pressure at which the authors observed isothermal entropy change anomalies, see Fig. 2 c and d. In the same way, it would be highly possible that the β - Li_3N polymorph also shows a phase transition induced by uniaxial strain. In this context, the authors should comment on the already reported pressure effects on Li_3N polymorphs and show further evidences for the absence of structural transitions that they are claiming in both baro- and elastocaloric effects.

(2) Following the previous comment, it would be also highly recommendable to perform experimental

analysis on mechanocaloric effects for the α -Li₃N and β -Li₃N polymorphs in order to confirm the values that the authors have theoretically obtained.

(3) It would be very useful for the audience if the authors can include a graphic of the diffusion coefficient and ionic conductivity as a function of pressure/strain.

(4) In addition to the mechanocaloric effects of Li₃N, the authors calculated the CO for different superionic materials, namely Li₃N, Li₃P and Li₃As. On this basis, they claim that Li₃P and Li₃As are better mechanocaloric materials than Li₃N. Therefore, why did they performed the theoretical analysis on Li₃N instead of doing it on Li₃P and Li₃As? The authors should clarify this point. Without the corresponding theoretical calculations on the mechanocaloric effects of Li₃P and Li₃As, it is not recommendable to claim that they are better mechanocaloric materials than Li₃N.

(5) The authors claim that biaxial stresses are likely to render similar mechanocaloric effects than obtained with uniaxial stresses. Nevertheless, they have not performed these biaxial simulations. In turn, it is not recommendable to do such claim without further evidences.

(6) The authors should update the references with the latest reported works on mechanocaloric effects: A. Corrales-Salazar et al., *Phys. Rev. Materials*, 2017, 1, 053601; J. M. Bermudez-Garcia et al., *J. Phys. Chem. Lett.*, 2017, 8 (18), 4419; P. J. von Ranke et al., *J. Alloys Compd.*, 2018, 749, 556.

(7) The authors should include a table comparing the mechanocaloric effects that they calculated for Li₃N with other (experimental and/or theoretical) values reported in the literature for other mechanocaloric materials.

=====
Reviewer 1
=====

The submitted paper describes a room-temperature mechanocaloric effects in lithium-based superionic materials. The mechanocaloric effects ($|\Delta S| \sim 25 \text{ JK}^{-1}\text{kg}^{-1}$ at a maximum hydrostatic stress of 1 GPa and $|\Delta T| \sim 5 \text{ K}$) of Li_3N are not remarkable compared with the rare-earth serial materials, however Li_3N has a distinctly different feature. As the fast-ion conductor, Li_3N has no structural phase transformation, so there is no hysteresis in the phase transformation which is fully reversible. This topic is interesting, and the presentation of the results appears to be clear and the paper fairly well written. Therefore, I recommend the paper should be published.

A: We are thankful to the Reviewer for his/her accurate appreciation of our work and positive recommendation on it.

=====
Reviewer 2
=====

The authors have reported a so called large mechanocaloric effects in the fast-ion conductor Li₃N which is absence of any structural phase transformation. The results sound interesting, but there are several issues still unclear.

A: We would like to thank the Reviewer for his/her evaluation of our work and positive comments on it.

1. The description of the measurement of the mechanocaloric effect in the manuscript make me confusing which is the base of the manuscript. I. e., the reliability of the temperature change, and the durable.

A: This is a theoretical work in which we employ well-established simulation techniques to estimate the presence and size of mechanocaloric effects in the room-temperature fast-ion conductor Li₃N. Our methods are carefully explained in the Methods and Supplementary Methods, and we also refer to previous works in which additional details can be found (e.g., Refs. [19-21] and [25-26] in the revised manuscript). It is not clear to us which specific part of our simulations and/or results are causing confusion to the Reviewer. Nevertheless, we have made great efforts at improving the readability of our work by adding new pieces of information on our simulation and analysis methods (see new Supplementary Figures, Supplementary Table, references, and clarifications, all highlighted in blue in the main text).

2. The pressure and temperature dependence of ΔT and ΔS should be provided.

A: In Figure 2, the Reviewer may find our ΔT and ΔS results corresponding to direct barocaloric effects expressed as a function of hydrostatic pressure (σ) and temperature (T). In Figure 5, we present our ΔT and ΔS results corresponding to inverse elastocaloric effects expressed as a function of temperature and uniaxial strain. In this latter case, we have chosen to represent our results as a function of uniaxial strain, rather than uniaxial stress, because in actual elastocaloric experiments uniaxial strain is the quantity that is externally adjusted; nevertheless, in the inset of Figure 5 we provide the relation between uniaxial strain and uniaxial stress in our simulations, hence our ΔT and ΔS elastocaloric results can be already worked out straightforwardly as a function of temperature and uniaxial stress.

3. The origin of such large MC effect should be discussed.

A: In the original version of our work we already discussed the origins of the large mechanocaloric effects disclosed in the room-temperature fast-ion conductor Li_3N , namely, stress-induced variations on the ionic conductivity of the material that in turn provoke large and reversible changes in the entropy and volume of the system (see, for instance, the Abstract, Introduction, Results, and Discussion sections). Nevertheless, in order to follow the Reviewer's recommendation, in the revised version of our article we have added Supplementary Figures 5 and 6, in which we explicitly show the effects of stress on the kinetic energy barriers governing lithium ion diffusion and on the corresponding ionic conductivity.

4. The discussion related to difference and relation between the materials with and without structure transformation are suggested.

A: Following the Reviewer's suggestion, in the revised version of our article we have added Supplementary Table 2, in which we compare the size and thermodynamic conditions of mechanocaloric effects occurring in Li_3N to those occurring in other materials that are originated by structural phase transformations.

5. Additionally, I do not really agree with the authors declared that, the sample is without structure transformation, such high pressure may at least induce the lattice distortion, it is better to provide more evidence.

A: We totally agree with the Reviewer in that application of pressure will always induce a lattice distortion in the system. However, our definition of “structural phase transition” stands for “a change in the crystal symmetry of the system” (for instance, a typical $\text{bcc} \rightarrow \text{fcc}$ or $\text{fcc} \rightarrow \text{hcp}$ phase transformation). In order to avoid possible misunderstandings, in the revised version of our article we have clarified that by structural phase transformation we mean a change on the symmetry of the crystal (see text highlighted in blue in page 2).

=====
Reviewer 3
=====

In the manuscript entitled “Room-temperature mechanocaloric effects in lithium-based superionic materials”, the authors present promising theoretical simulations on giant barocaloric effects in the Li₃N superionic materials, specifically in the α -Li₃N and β -Li₃N polymorphs, at 300 and 400 K. This is a very interesting and well-written work on the growing topic of mechanocaloric materials. The more striking features of the here exposed manuscript is the calculation of giant mechanocaloric effects in the absence of any structural transition found in two Li₃N polymorphs.

A: We would like to thank the Reviewer for his/her careful assessment of our work and positive evaluation. Also, for his/her insightful comments, which have allowed us to improve our manuscript substantially.

Nevertheless, I found several major concerns that authors should address previous publication of this work:

1. According to the authors, the giant mechanocaloric effects showed by α -Li₃N and β -Li₃N polymorphs are not driven by any structural transition. Nevertheless, there are several publications reporting several pressure-induced phase transitions in Li₃N [S. Cui et al., Solid State Commun., 2009, 149, 612; Y. Yan et al., Eur. Phys. J. B, 2008, 61, 397; J. C. Schön et al., J. Mater. Chem., 2001, 11, 69; M. Mali et al., Phys. Rev. B, 1987, 36 (7), 3888; A. C. Ho et al., Phys. Rev. B, 1999, 59 (9), 6083]. Even more, it is reported that at around 0.42 GPa there is a pressure-induced phase transition from α -Li₃N to β -Li₃N [W. Li et al., Energy Environ. Sci., 2010, 3, 1524], pressure at which the authors observed isothermal entropy change anomalies, see Fig. 2 c and d. In the same way, it would be highly possible that the β -Li₃N polymorph also shows a phase transition induced by uniaxial strain. In this context, the authors should comment on the already reported pressure effects on Li₃N polymorphs and show further evidences for the absence of structural transitions that they are claiming in both baro- and elastocaloric effects.

A: The latest experimental and theoretical reports mentioned by the Reviewer (i.e, S. Cui et al., Sol. Stat. Commun., 2009, 149, 612; Y. Yan et al., Eur. Phys. J. B, 2008, 61, 397; A. C. Ho et al., Phys. Rev. B, 1999, 59, 6083) consistently agree in that no phase transition occurs in bulk Li₃N as driven by hydrostatic pressures lying within the interval $0 < P < 1$ GPa. In particular, the reported transition pressures for the transformation between the alpha and beta Li₃N polymorphs are around 1.5 GPa in the theoretical works by Cui and Yan, and in Ho's experimental study it is stated that “Upon loading, the P6/mmm structure was completely transformed to the P63/mmc structure. The P6/mmm structure was not observed at or above the lowest pressure of

3 GPa.” (second column in second page). These previous theoretical and experimental results are consistent with our observations that no pressure-induced phase transformation occurs during our simulations covering the pressure interval $0 < P < 1$ GPa. This is clarified in the new version of our manuscript (second column, second page), and the works mentioned above now appear cited therein.

We should note that there seems to be a mistake in the Reviewer's statement: “*Even more, it is reported that at around 0.42 GPa there is a pressure-induced phase transition from α -Li₃N to β -Li₃N [W. Li et al., Energy Environ. Sci., 2010, 3, 1524]...*”; the reported transition pressure in that reference is actually of 4.2 kPa (that is, 0.0000042 GPa, see second column in first page), which does not coincide with any anomaly in our estimations and comes to show that both the alpha and beta Li₃N polymorphs in fact can be realized at normal conditions depending on the employed synthesis method (the latter polymorph as a metastable state).

We have followed the Reviewer's advice and provided additional evidence showing the absence of any stress-induced structural phase transformation in our simulations covering the hydrostatic pressures of $0 < P < 1$ GPa and uniaxial tensile stresses of $0 < |\sigma| < 9$ GPa. In particular, we have added the following Supplementary Figures, briefly explained next, to which we refer to and comment throughout the main text in our revised manuscript:

- Supplementary Figure 2: shows the coordination numbers of Li-Li and N-N ionic couples calculated along molecular dynamics simulations performed at different mechanical stress conditions and room temperature; no variations are appreciated as compared to the values obtained at zero pressure.

- Supplementary Figure 3: shows the position correlation function of Li and N species (defined in the caption of the figure and in Refs. [8,9] of the Supplementary Information) calculated along molecular dynamics simulations performed under different pressure conditions and at room temperature; no permanent vibrational displacements are acquired by the N atoms at high pressure.

- Supplementary Figure 4: first-principles enthalpy and energy calculations showing that no phase transformation occurs among the alpha and beta Li₃N polymorphs for hydrostatic pressures of $0 < P < 1$ GPa or uniaxial tensile stresses of $0 < |\sigma| < 9$ GPa.

2. Following the previous comment, it would be also highly recommendable to perform experimental analysis on mechanocaloric effects for the α -Li₃N and β -Li₃N polymorphs in order to confirm the values that the authors have theoretically obtained.

A: Unfortunately, our research expertise is mostly on theoretical methods and we do not have any experience on measuring caloric effects in materials. We hope that our

simulation results will actually stimulate experimental searches of large mechanocaloric effects in room-temperature superionic materials. We would like to mention here a previous theoretical work of ours on caloric effects in superionic AgI (Nat. Commun. 8, 963 -2017-) that actually motivated a quick and successful experimental validation (Nat. Commun. 8, 1851 -2017-) by the experimental research groups specialized in caloric materials at the University of Cambridge (lead by Profs. Moya and Marthur), University of Barcelona (lead by Profs. Mañosa and Planes), and Polytechnic University of Catalonia (lead by Prof. Tamarit).

3. It would be very useful for the audience if the authors can include a graphic of the diffusion coefficient and ionic conductivity as a function of pressure/strain.

A: Following the Reviewer's suggestion, that piece of information appears now in the revised version of our article in Supplementary Figure 6.

4. In addition to the mechanocaloric effects of Li₃N, the authors calculated the C₀ for different superionic materials, namely Li₃N, Li₃P and Li₃As. On this basis, they claim that Li₃P and Li₃As are better mechanocaloric materials than Li₃N. Therefore, why did they performed the theoretical analysis on Li₃N instead of doing it on Li₃P and Li₃As? The authors should clarify this point. Without the corresponding theoretical calculations on the mechanocaloric effects of Li₃P and Li₃As, it is not recommendable to claim that they are better mechanocaloric materials than Li₃N.

A: Our claim that Li₃P and Li₃As could present larger mechanocaloric effects than Li₃N is based on physically sound arguments. We would like first to recall here two of the main formulas involved in our theoretical estimations of ΔT and ΔS , which already appear in the manuscript (Eqs. 3 and 5 in Methods), that read as:

$$\Delta S(\sigma_f, T) = - \int_0^{\sigma_f} \left(\frac{\partial V}{\partial T} \right)_{\sigma} d\sigma \quad (1)$$

and

$$\Delta T(\sigma_f, T) = - \frac{T}{C_0(T)} \cdot \Delta S(\sigma_f, T). \quad (2)$$

It is experimentally known that Li₃P and Li₃As are better fast-ion conductors than Li₃N at room and higher temperatures (see Ref. [24] in the revised version of our article), and that all these Li₃X compounds are isostructural. This means that the changes of entropy as induced by pressure in Li₃P and Li₃As may be expected to be at least as large as in Li₃N (see new reference [35] in the revised version of our manuscript, where that is explained as “...In the case of lithium/vacancy orderings that only differ in the arrangement of lithium ions and vacancies on their respective

sublattice, the largest contribution to entropy can be expected to be due to the configurational degrees of freedom. Although the magnitude of the vibrational entropy may be significant, its variations across phases of the same chemical species are typically small so that it does not much affect the relative phase stability...” second column, fourth page). Actually, we have found by means of first-principles calculations based on the quasi-harmonic approximation that the volume expansion coefficient dV/dT appearing in the integrand of Eq.(1) above, is significantly larger in Li_3As than in Li_3N at ambient conditions (see new Supplementary Figure 10); therefore, it is highly plausible to expect that the adiabatic entropy change ΔS achieved in Li_3As and Li_3P will be larger (or at least of the same order of magnitude) than in Li_3N . In addition to this finding, we have demonstrated that the heat capacity C_0 of Li_3As and Li_3P are significantly smaller than that of Li_3N (Supplementary Figure 1); consequently, upon consideration of Eq.(2) above, where ΔS appears in the numerator and C_0 in the denominator, it is reasonable to expect that larger ΔT can be achieved in Li_3As and Li_3P than in Li_3N . This type of reasoning now appears more elaborated and better supported (i.e., new Ref.[35] and new results reported in Supplementary Figure 10) in the revised version of our manuscript.

The reason why we have analyzed Li_3N instead of Li_3As or Li_3P is of purely technical nature: We have not found any interatomic potential model in the literature describing the thermodynamic stability and superionic properties of Li_3As or Li_3P in a physically reliable manner. The availability of reliable interatomic potential models represents a critical ingredient in our modelling work, as our molecular dynamics simulations and the accuracy in our ΔT and ΔS estimations rely entirely upon them. Unfortunately, reliable interatomic potential models for fast-ion conductors Li_3As and Li_3P appear to be missing in the literature. We even tried to devise some Born-Mayer-Huggings potentials for Li_3As by ourselves, but did not succeed in our attempts. Nevertheless, it must be noted that the main conclusions and the reported atomistic mechanisms supporting large mechanocaloric responses in room-temperature lithium-based superionic materials, remain being totally valid and general.

5. The authors claim that biaxial stresses are likely to render similar mechanocaloric effects than obtained with uniaxial stresses. Nevertheless, they have not performed these biaxial simulations. In turn, it is not recommendable to do such claim without further evidences.

A: In view of the Reviewer's point, we have performed new molecular dynamics simulations in which we have explicitly assessed the mechanocaloric effects in Li_3N resulting from applying compressive biaxial stresses. Those new results appear in the Supplementary Figure 11, and confirm our initial hypothesis (which was reasonably based in the outcomes of our previous work [20]). In the revised version of our manuscript, therefore, we have kept such a claim.

6. The authors should update the references with the latest reported works on mechanocaloric effects: A. Corrales-Salazar et al., Phys. Rev. Materials, 2017, 1, 053601; J. M. Bermudez-Garcia et al., J. Phys. Chem. Lett., 2017, 8 (18), 4419; P. J. von Ranke et al., J. Alloys Compd., 2018, 749, 556.

A: Following the Reviewer's advice, all the references mentioned above appear now in the revised version of our article.

7. The authors should include a table comparing the mechanocaloric effects that they calculated for Li₃N with other (experimental and/or theoretical) values reported in the literature for other mechanocaloric materials.

A: Following the Reviewer's advice, such a table now appears in the revised version of our manuscript (see Supplementary Table 2).

Reviewers' comments:

Reviewer #2 (Remarks to the Author):

The authors have improved the manuscript accordingly. However, I still have some doubt about the discussion, especially for the only theoretical predication. Moreover, the related theory and experiment investigation (without the so-called large mechanocaloric effects) of the same compounds have been published even more than 10 years (S.Cui et al., Sol. Stat. Commun., 2009, 149, 612; Y. Yan et al., Eur. Phys. J. B, 2008, 61, 397; A. C. Ho et al., Phys. Rev. B, 1999, 59, 6083, W. Li et al., Energy Environ. Sci., 2010, 3, 1524). I can not recommend the manuscript for publication at Nature Communication. I would like to reconsider my decision only if the authors can provide some experiment evidence to support the results.

Reviewer #3 (Remarks to the Author):

The authors have succesfully addressed all my comments. Therefore, I recommend this work for publication in Nature Communications without further modifications.

=====
Reviewer 2
=====

The authors have improved the manuscript accordingly.

A: We are glad to acknowledge that the Reviewer appreciates our efforts in trying to address his/her criticisms.

However, I still have some doubt about the discussion, especially for the only theoretically predication.

A: In order to avoid possible misunderstandings, we have rewritten the Discussion section in a more appropriate and clear way (see the two first paragraph in that section in our revised manuscript). We hope that the revised version of the Discussion section will be now more understandable.

Moreover, the related theory and experiment investigation (without the so-called large mechanocaloric effects) of the same compounds have been published even more than 10 years (S.Cui et al., Sol. Stat. Commun., 2009, 149, 612; Y. Yan et al., Eur. Phys. J. B, 2008, 61, 397; A. C. Ho et al., Phys. Rev. B, 1999, 59, 6083, W. Li et al., Energy Environ. Sci., 2010, 3, 1524).

A: Certainly, Li_3N is a very well-know compound and there are considerable amounts of work, both experimental and theoretical, done in this material (as we illustrate in the Bibliography section of our manuscript). Yet, the main point in our article is not the characterization of the phase diagram of Li_3N . Our main finding is that Li_3N presents large and dual mechanocaloric effects at room temperature in the absence of any structural phase transition. This is in fact a very novel result that has the potential to improve the performance of solid-state refrigeration cycles, since it provides a simple strategy to avoid the usual and detrimental mechanical hysteresis effects affecting them. Meanwhile, the mechanocaloric effects that we have revealed in Li_3N are very likely to exist also in a number of diverse and abundant fast-ion conductors (e.g., lithium-based - LiFePO_4 - and solid-oxide fuel cell - $\text{La}_x\text{Sr}_{1-x}\text{MnO}_3$ - materials). In this sense, Li_3N represents a “proof of concept” that opens new and exciting possibilities in the field of solid-state cooling by taking advantage of what is already known from electrochemical devices (that is, knowledge transfer).

We note in passing that the fact that Li_3N is a very well know material has permitted us to assess the reliability of our computational approach against experiments (and also against previous calculations). In particular, we have demonstrated very good agreement with respect to measurements performed on the structural, energy, and ion-transport properties of this material. In the new version of our manuscript, we have

stressed the consistent agreement between our simulations and previous experimental results in Li_3N , which comes to reassure the validity of our computational approach.

I can not recommend the manuscript for publication at Nature Communication. I would like to reconsider my decision only if the authors can provide some experiment evidence to support the results.

A: Unfortunately, our research expertise is on theoretical and computational methods and we do not have experience in measuring caloric effects. We believe that our simulation results will stimulate new experimental searches of large room-temperature mechanocaloric effects by specialized groups, and that in general they will attract great attention within the community of solid-state cooling research.

We understand that the Reviewer has an experimental background and that our approach based on simulation and theory probably falls a bit out of his/her expertise field. Nevertheless, we would like him/her to consider the following points in order to make a fair evaluation of our work.

1. Computational approaches are reliable and have predictive power

We would like to mention here a previous theoretical work of ours in which we predicted the existence of giant caloric effects in AgI (Nat. Commun. 8, 963 -2017-) by using an analogous computational approach than employed here. Our article motivated a quick and successful experimental validation (namely, Nat. Commun. 8, 1851 -2017-) by the experimental research groups specialized in caloric materials at the University of Cambridge (led by Profs. Moya and Marthur) and University of Barcelona (led by Profs. Mañosa and Planes). This example very well illustrates the scientific rigor and reliability of our theoretical findings and analysis. Analogous examples abound in the literature (see new references in the revised manuscript).

2. Novel and technically sound predictions deserve being published in high-impact journals independently of whether they are accompanied by experiments or not

Certainly, our work is a theoretical investigation (based on well-tested first-principles and molecular dynamics methods, let us stress) and no experimental validations are provided in the manuscript. Yet our findings are highly original and paradigm-changing, which very well fits to a journal like Nature Communications and its readership. Actually, Reviewers 1 and 3, who presumably are experimentalists as well, have come to agree to this point. It is worth noting here that in many scientific fields theoretical works guide the best experimental efforts, not necessarily performed within a same research group, and that as a result new transformative technologies emerge. This is the modern approach that, for instance, various government agencies in the United States, Europe, China and other countries are pursuing with initiatives like the “Materials Genome Initiative”, “Integrated Computational Materials Engineering”, and “Big Data”. We firmly believe that our work belongs to such a class of transformative theoretical research; therefore, objecting publication of our

work because of the lack of laboratory validation certainly is hidebound and unfair.

In the revised version of our manuscript, we have stressed point 1 as explained above both in the main text (see sentences highlighted in blue therein and the two additional references) and Supplementary Materials (see last paragraph in the “Supplementary Methods” section and new caption of Supplementary Fig.7).

=====
Reviewer 3
=====

The authors have successfully addressed all my comments. Therefore, I recommend this work for publication in Nature Communications without further modifications.

A: We would like to thank the Reviewer for his/her careful evaluation of our work and positive final recommendation on it.